∂ | Open Peer Review | Veterinary Microbiology | Research Article

# Development and optimization of a double antigen sandwich ELISA detecting for Senecavirus A antibodies based on VP2 protein

Jie Chen,[1,2] Zhengwang Shi,[1,2] Juncong Luo,[1,2] Caixia Jia,[1,2] Xiaoyang Zhang,[1,2] Juanjuan Wei,[1,2] Shuaipeng Li,[1,2] Yuqian Zhu,[1,2] Tao Xi,[1,2] Jing Zhou,[1,2] Yindi He,[1,2] Xintai Shi,[1,2] Huanchen Liao,[1,2] Hong Tian,[1,2] Haixue Zheng[1,2]

**ABSTRACT**  Senecavirus A (SVA) is an emerging viral pathogen that threatens the global swine industry significantly. The major clinical symptoms of SVA-infected animals are vesicular lesions, but various diseases can cause the same symptoms, which makes it difficult to distinguish SVA from other vesicular diseases clinically. The absence of an effective and safe vaccine necessitates the development of a simple, specific, and sensitive serological detection method for SVA antibodies. The VP2 protein of SVA, characterized by high immunogenicity and sequence conservatism, is an essential target for serological diagnosis. In this study, a double-antigen sandwich enzyme-linked immunosorbent assay [ELISA (DAgS-ELISA)] based on VP2 protein expressed by *Escherichia coli* was established for SVA antibody detection. With a cut-off value of 0.237, this assay demonstrated outstanding performance, showing high sensitivity and sharp specificity, which is manifested in the absence of cross-reaction with classical swine fever virus (CSFV), African swine fever virus (ASFV), pseudorabies virus (PRV), and porcine reproductive and respiratory syndrome virus (PRRSV), and foot-and-mouth disease virus (FMDV) serotype A and O. Additionally, the repeatability of the method is remarkable, as shown by the coefficients variation (CV) of both the intra- and inter-assay below 10%. By detecting 166 clinical sera, it was found that the kappa value of the DAgS-ELISA was 0.78 compared with that of the virus neutralization test (VNT), indicating a high level of consistency. In general, this method has high sensitivity, sharp specificity, remarkable repeatability, sound consistency, and low cost, making it a reliable and effective tool for detecting SVA antibodies.

**IMPORTANCE**  SVA has rapidly become prevalent in many countries, and its outbreaks have threatened the global swine industry significantly. The major clinical symptoms of SVA-infected animals are vesicular lesions that are similar to other vesicular diseases, making it difficult to distinguish SVA. Currently, no commercial vaccines are available for SVA; therefore, effective diagnosis of SVA infection is vital for its prevention and control. In this study, VP2 protein of SVA was expressed by *E. coli*, and a double-antigen sandwich enzyme-linked immunosorbent assay [ELISA (DAgS-ELISA)] for SVA antibodies detection was successfully established based on the VP2 protein. The DAgS-ELISA has a high sensitivity, sharp specificity, remarkable repeatability, sound consistency, and low cost for detecting SVA antibodies. Therefore, the DAgS-ELISA established in this study may be a reliable and effective tool for detecting SVA antibodies and may be used to strengthen the monitoring and prevention of SVA epidemic in the long run.

**KEYWORDS**  Senecavirus A, VP2 protein, antibodies, double-antigen sandwich enzyme-linked immunosorbent assay (DAgS-ELISA)

**Peer Reviewer** Kaan Çeylan, Faculty of Medicine University of Gaziantep, Gaziantep, Turkey

Address correspondence to Hong Tian, xibeitian0931@163.com, or Haixue Zheng, zhenghaixue@caas.cn.

The authors declare no conflict of interest.

See the funding table on p. 11.

*[This article was published on 22 October 2024 with an error in the figures. The correct figures were uploaded in the current version, posted on 13 November 2024.]*

Senecavirus A (SVA), previously known as Seneca Valley virus (SVV), is the only member of the *Picornaviridae* family and *Seneca virus* genus (1–3) and is one of the pathogenic agents associated with the swine idiopathic vesicular disease (SIVD) (2). In 2002, SVA was initially discovered in a laboratory in Maryland, thought to be an unknown foreign contaminant in cell culture and could be filtered through 0.22 µm, then the prototype strain, named SVV-001, was isolated from the pathogenic contaminant in human embryonic retinal cell PER.C6 culture (4). The contaminant was thought to have originated from bovine serum in the culture medium or porcine trypsin used to digest cells (5).

SVA is a single-stranded unenveloped RNA virus with an icosahedral virion about 30 nm in diameter and 7.2 kilobases in genome size (4). A 5′ untranslated region (UTR), an open reading frame, and a 3′-UTR comprise its genome, and the structural proteins named VP4, VP2, VP3, and VP1 are encoded by the P1 region. During assembly, the P1 polypeptide is initially cleaved by 3C protease to produce VP0, VP3, and VP1. As VP0 matures, it is cleaved to generate VP2 and the internally localized VP4, which together form viral capsids; meanwhile, seven non-structural proteins are encoded by the P2 and P3 regions (6). Among all structural proteins, VP2 is an ideal diagnostic target for specifically detecting SVA antibodies due to its good immunogenicity and the ability to induce neutralizing antibodies (7, 8). In our previous studies on the screening and identification of coated antigens, VP2 depicts an obvious advantage in terms of specificity and sensitivity compared with VP1 and VP3 (data not presented), which is consistent with the findings.

In 2014, reports of SVA infections were made in the United States, Canada, Brazil, and Columbus (5). Subsequently, SVA outbreaks occurred successively in more countries and gradually spread, including China (9, 10), Thailand (11), Vietnam (12), Colombia (13), and Chile (14). In 2015, SVA infection was first identified in swine with symptoms of vesiculosis in Guangdong and then rapidly spread to several provinces in China (15–21), which indicates that SVA epidemic areas are not only widely distributed but also have a trend of further spread. The clinical symptoms of SVA infection are very similar to those of FMDV, vesicular stomatitis virus, swine vesicular disease virus, and vesicular exanthema of swine virus, which increases the difficulties for clinical diagnosis (5, 22, 23). Meanwhile, SVA has been recognized as an endemic syndrome that negatively affects the swine industry worldwide (20, 24–27). Currently, no commercial vaccines are available for SVA, and the detection of SVA antibodies suggests that the body has been or is in the process of infection. For these reasons, effective diagnosis of SVA infection is vital for its prevention and control.

To date, a variety of diagnostic methods are available to detect SVA infection in pigs, including the virus neutralization test (VNT), indirect immunofluorescence assay (IFA), polymerase chain reaction (PCR) (27–29), indirect enzyme-linked immunosorbent assay (ELISA) (30), and competitive ELISA (24, 31–36). Among them, VNT is considered to be the golden criterion for detecting antibodies in animal serum (37), according to the recommendations of the World Organization for Animal Health. However, VNT and IFA take too long (typically require 48–72 h) and are operationally complex, making them less convenient for clinical sample detection. The main drawback of PCR lies in that its sensitivity depends upon the timing of appropriate sample collection, that is, PCR has a good detection effect only when the animal is in the state of acute infection, but it is difficult to detect by PCR when the animal is recovered or infected with attenuated strains, although it is very sensitive and specific in diagnosis of infectious diseases (38). Compared with the methods mentioned above, ELISA and other serological methods are widely used because they are fast and inexpensive, but there are still limitations such as window period and cross-reaction issues (39). Therefore, it is essential to establish a simple and economical ELISA method with excellent performance for the detection of SVA antibodies.

In view of the increasing prevalence of SVA at home and abroad, this study aims to develop and characterize a double-antigen sandwich ELISA (DAgS-ELISA) for detecting

SVA antibodies. To our knowledge, this is also the first establishment of DAgS-ELISA that can be used for the identification of SVA as well as the differential diagnosis of SIVD caused by SVA, which will lay the foundation for the prevention and control of SVA in the long run.

## RESULTS

### Expression, purification, and digestion of VP2 protein

According to SDS-PAGE, it was found that only a minority was present in the supernatant. Concurrently, a large portion is located in the precipitate (Fig. 1A) and has a molecular weight of approximately 37 kDa. After the digestion through thrombin, the molecular weight of VP2 protein turned into 35 kDa (Fig. 1B) as expected.

### Reaction of purified VP2 protein

The indirect ELISA results demonstrated that the VP2 protein reacted with SVA-positive serum but not with SVA-negative serum (Fig. 2A). Similarly, western blotting analysis revealed that when detected with SVA-positive serum, VP2 protein showed a distinct band at approximately 35 kDa but was not responsive to SVA-negative serum (Fig. 2B), which indicates the VP2 protein is of high reactivity. The VP2 protein with His tag exhibited a distinct band at about 37 kDa, but after digestion with thrombin, it did not react with the His tag antibody (Fig. 2C), demonstrating that the His tag of VP2 protein had been successfully removed.

### Optimization of the DAgS-ELISA conditions

In the checkerboard titration testing, an optimized concentration of the coated antigen in 1 µg/mL (equivalent to 50 ng/well), as well as a serum dilution ratio of 1:2 was selected because of a maximum P/N value of 15.969 (Table 1). In this case, the optimal serum

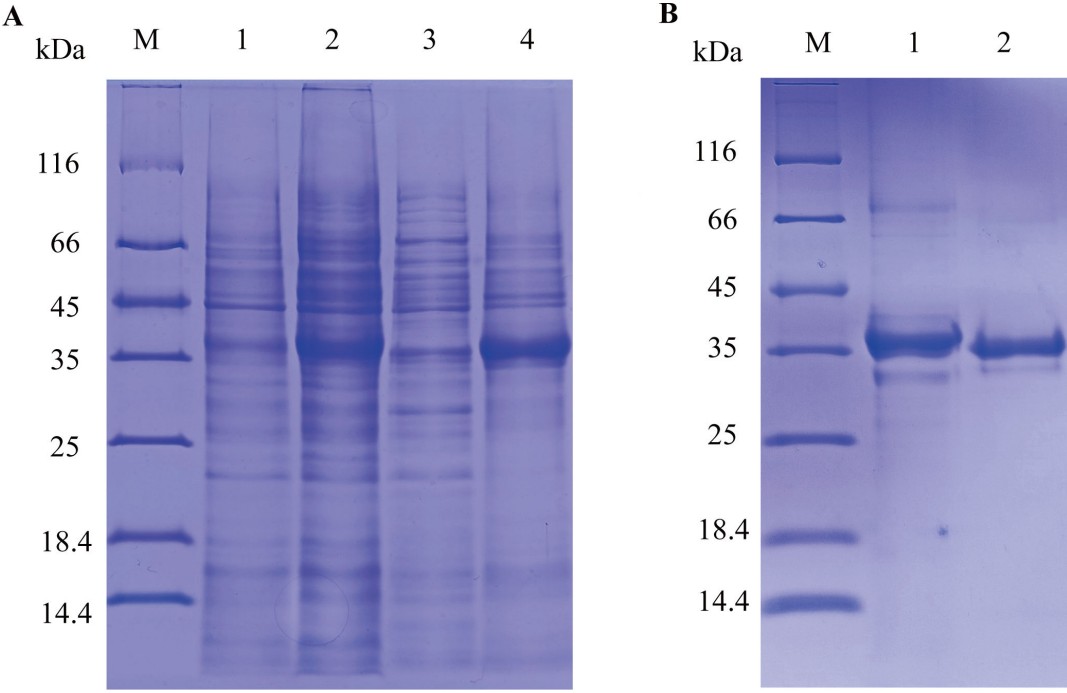

**FIG 1** Expression and purification of the VP2 protein. (A) (M) Protein molecular weight standard (1) uninduced pET-30a-VP2 bacteria; (2) induced pET-30a-VP2 bacteria; (3) supernatant of sonicated lysate of induced pET-30a-VP2 bacteria; and (4) precipitant of sonicated lysate of induced pET-30a-VP2 bacteria. (B) Removing the His tag of VP2 protein through thrombin. (M) Protein molecular quality standard: (1) the VP2 protein with His tag; (2) the VP2 protein without His tag.

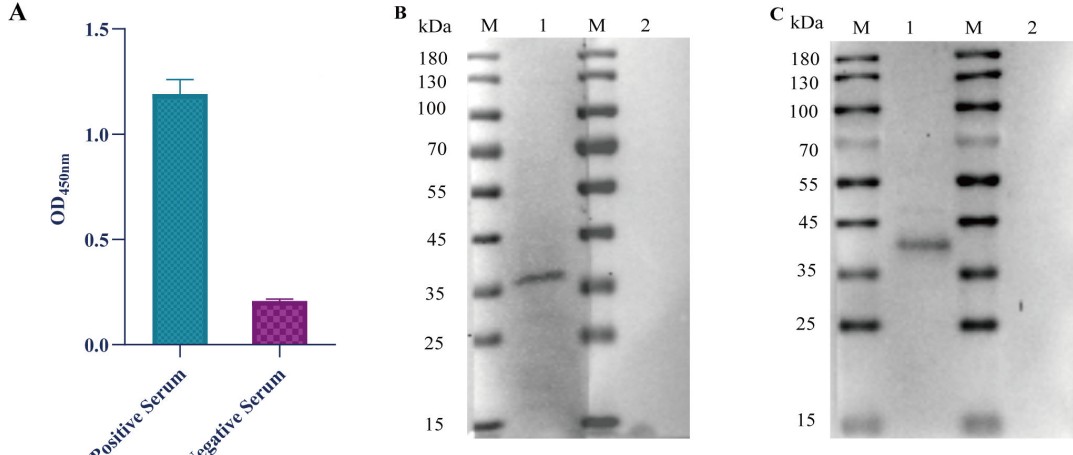

**FIG 2** Reaction of purified VP2 protein. (A) Indirect ELISA results of VP2 protein and SVA serum. SVA-positive/negative serum (1:10) was used as the primary antibody, and HRP-labeled goat anti-pig IgG (1:30,000) as the second antibody. (B) Western blotting analysis of purified VP2 protein and SVA serum. SVA-positive/negative serum (1:1,000) was used as the primary antibody, and HRP-labeled goat anti-pig IgG (1:10,000) was used as the second antibody. (M) protein marker: (1) SVA positive serum; (2) SVA negative serum. (C) Western blotting analysis of VP2 protein before and after thrombin digestion. mAb to 6 × His tag (1:5,000) was used as the primary antibody, and HRP-labeled rabbit anti-mouse IgG (1:15,000) as the second antibody. (M) Protein marker (1); VP2 protein with His tag (2); VP2 protein without His tag.

reaction time, dilution ratio, and reaction time of HRP-labeled antigen were sequentially optimized based on the P/N value, followed by optimizing the TMB reaction time. After a series of screening and optimization, the optimal working conditions for the DAgS-ELISA were determined as incubating with serum for 60 min (Fig. 3A), diluting the HRP-labeled antigen at a ratio of 1:80,000 (Fig. 3B) and incubating for 30 min (Fig. 3C), followed by reacting with TMB for 15 min (Fig. 3D).

## Cut-off value of DAgS-ELISA

Totally, 130 SVA-negative serum samples were detected under the optimized conditions, with an average ($\overline{X}$) of the $OD_{450nm}$ values of 0.125 and an SD of 0.037, leading to a cut-off value of ($\overline{X}$ +3 SD) = 0.237 (Fig. 4).

## Sensitivity and specificity of DAgS-ELISA

The optimized DAgS-ELISA was used to detect SVA standard serum and that of common swine pathogens, three samples for each. Except for SVA-positive serum, the $OD_{450nm}$ value of other samples was far below the cut-off value (Fig. 5A), demonstrating that the

**TABLE 1** Determination of optimal antigen coating concentration and serum dilutions[a]

| Dilution of sera | | Antigen at different concentrations (µg/mL) | | | | |
|---|---|---|---|---|---|---|
| | | 0.5 | 1 | 2 | 4 | 8 |
| 1:2 | P | 2.537 | 2.603 | 1.919 | 1.360 | 1.240 |
| | N | 0.174 | 0.163 | 0.184 | 0.199 | 0.218 |
| | P/N | 14.580 | *15.969* | 10.429 | 6.834 | 5.688 |
| 1:5 | P | 1.941 | 1.728 | 1.342 | 0.848 | 0.751 |
| | N | 0.124 | 0.179 | 0.108 | 0.136 | 0.170 |
| | P/N | 15.653 | 9.654 | 12.426 | 6.235 | 4.418 |
| 1:10 | P | 1.456 | 1.222 | 0.865 | 0.558 | 0.484 |
| | N | 0.106 | 0.121 | 0.118 | 0.140 | 0.149 |
| | P/N | 13.736 | 10.099 | 7.331 | 3.986 | 3.248 |

[a]The values depicted in bold italics indicate the values under the optimal conditions chosen for subsequent DAgS-ELISA. P: $OD_{450nm}$ value of positive samples; N: $OD_{450m}$ value of negative samples.

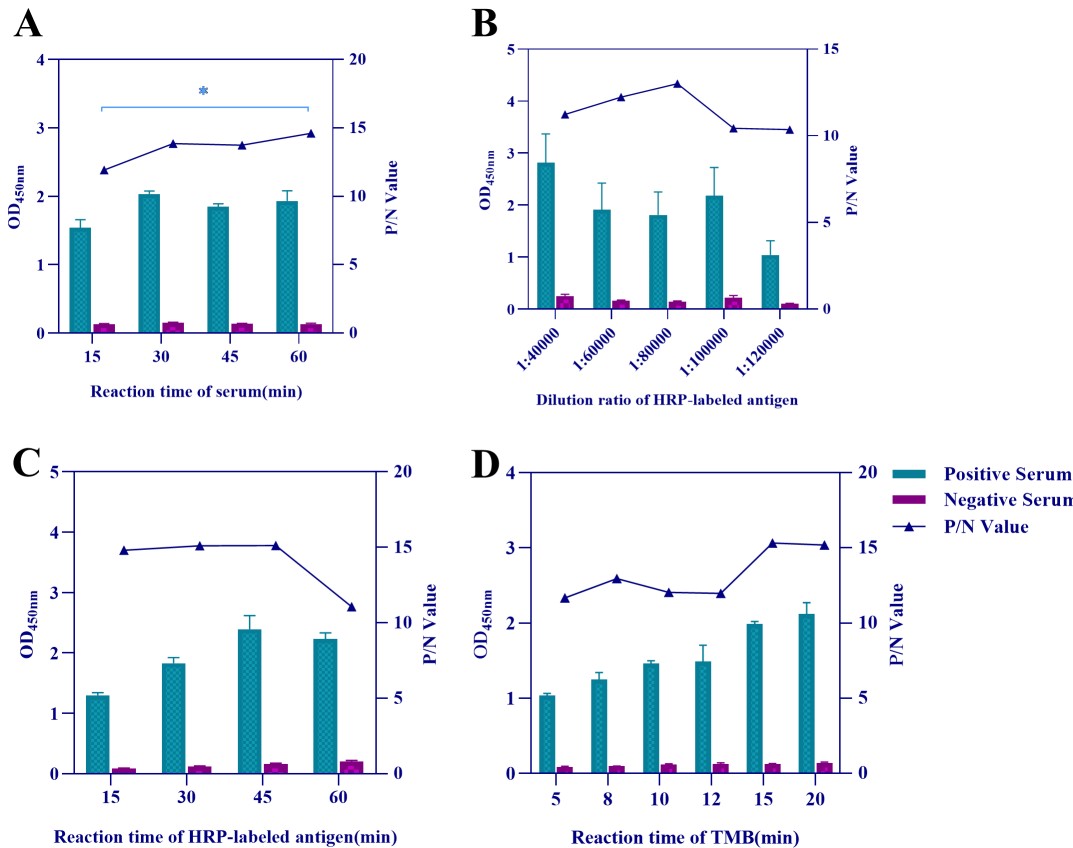

**FIG 3** Optimization results for the DAgS-ELISA. (A) Optimal reaction time of serum; (B) optimal dilution ratio of HRP-labeled antigen; (C) optimal reaction time for HRP-labeled antigen; and (D) optimal reaction time of TMB.

method was sharply specific. Besides, after a series of dilutions, the minimum detection limit of the method was 1:512 (Fig. 5B). The results above indicated that the DAgS-ELISA was a reliable method for detecting SVA antibodies with high specificity and sensitivity.

## Repeatability of DAgS-ELISA

The intra- and inter-assays were conducted to assess the repeatability of the method. CV is the ratio of SD to the average ($\overline{X}$), reflecting the degree of data dispersion to a certain extent, and a low CV generally signifies that the method is repeatable. As depicted in Table 2, the intra-assay CV ranges of the 10 serum samples were 1.77% to 7.00%, with a median of 4.78%, whereas the inter-assay CV of these samples was all below 10%. The above results suggested that the established DAgS-ELISA had the characteristics of high repeatability.

## Clinical serum sample detection

A total of 166 clinical swine serum samples were detected by the DAgS-ELISA and VNT. As illustrated in Table 3, 19 sera were positive, and 147 sera were negative in VNT test, whereas 28 were positive and 138 were negative in DAgS-ELISA. Of the same 166 serum samples, there are only nine tested positive for DAgS-ELISA but negative for VNT. The kappa value was calculated to be 0.78, and it can be indicated that the DAgS-ELISA established here has high agreement with that of VNT.

## DISCUSSION

As of July 2024, SVA has rapidly become prevalent in many countries, and its outbreaks have affected more than half of the Chinese provinces. According to the phylogenetic

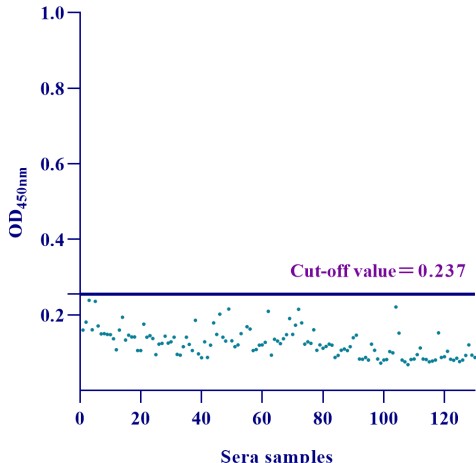

**FIG 4** Determination of the cut-off value of the DAgS-ELISA ($\overline{X}$ = 0.125, SD = 0.037, Cut-off value = 0.237).

analysis, SVA isolates in China could be mainly classified into two clusters and five genetically distinct branches (35). The SVA-associated vesicular diseases, including FMD, have caused great economic losses to the pig industry. Consequently, the establishment of comprehensive and systematic SVA serological diagnostic techniques is crucial to strengthen the monitoring and prevention of SVA epidemic.

Various types of antibodies in the sample, including IgM, IgA, and IgG, can be detected by DAgS-ELISA (40–42); Accordingly, the diseases can be detected in the early stages of infection. The method has been applied to human and animal diseases during pandemics, such as severe acute respiratory syndrome-associated coronavirus (43), coronavirus disease 2019, Chikungunya virus (44), and ASFV (45). The screening and identification of well-characterized rabbit polyclonal antisera or murine monoclonal antibodies is essential for indirect and competitive ELISA, which requires considerable time and effort. In comparison, the advantages of DAgS-ELISA are obvious: only antigens with higher purity and activity need to be prepared, which significantly shortens the production cycle and reduces costs. Furthermore, after the removal of the His Tag, interference of His Tag antibodies was avoided, and the specificity of the method was enhanced.

Although the detection results of the DAgS-ELISA method developed in this study and the VNT method were not perfectly 100% consistent (that is, among 166 clinical samples, 9 samples were positive for DAgS-ELISA but negative for VNT), the kappa value obtained by this calculation was only 0.78. However, from the essence of VNT test, it has the characteristics of high specificity and relatively low sensitivity, as only neutralizing antibodies can be detected by VNT, whereas all SVA-specific antibodies can be captured by the DAgS-ELISA we developed. This is also the reason why the kappa value between the two methods is not as high as expected. From another perspective, it shows that our DAgS-ELISA method has higher sensitivity and potential advantages over VNT.

Previous studies have indicated that the recombinant VP2 protein depicted good immunoreactivity and is an effective diagnostic target for establishing clinical diagnosis methods of SVA (33, 46). Generally, the prokaryotic expression system has shorter expression cycles and can effectively save costs. Therefore, the prokaryotic (*E. coli*) expression system was selected in this study, and the recombinant VP2 protein with strong immunogenicity and reactivity was successfully prepared. Based on this, the DAgS-ELISA method with high specificity, sensitivity, and reproducibility was further developed. For the specifically, it has no cross-reaction with CSFV, ASFV, PRV, PRRSV, and FMDV A/O and can be detected at a maximum dilution of 1:512, with a CV of the intra- and inter-assay below 10%. Compared with the VNT methods, it reveals excellent performance as well as a rapid and simple operation and can be better used for the

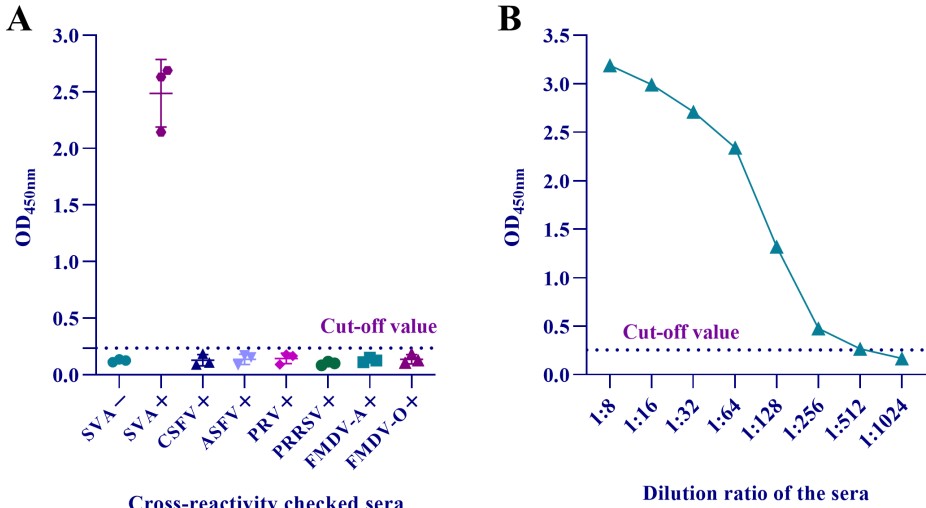

**FIG 5** Specificity and sensitivity of the DAgS-ELISA. (A) Specificity assay. SVA-negative serum, SVA-positive serum, and five other porcine pathogens-positive serum samples, including CSFV, ASFV, PRV, PRRSV, and FMDV A/O, were measured. Data represent ($\overline{X}$) ± SD from three independent experiments. (B) Sensitivity assay using the DAgS-ELISA. The dashed line represents the cut-off value.

detection of clinical serum samples. The DAgS-ELISA method established in this study has a comparative advantage because of its lower cost and shorter production cycles. It will establish a solid foundation for the further development of rapid and effective SVA antibody detection kits and serve as a model for the detection of other infectious diseases. Furthermore, it can be utilized to evaluate the herd immunity induced by various SVA-inactivated vaccines, so as to effectively prevent, control, and eradicate SVA.

## MATERIALS AND METHODS

### Serum samples and key materials

SVA, CSFV, ASFV, PRV, as well as porcine reproductive and respiratory syndrome virus (PRRSV)-positive serum samples were acquired from Lanzhou Veterinary Research Institute, Chinese Academy of Agricultural Sciences. Positive serum samples for foot-and-mouth disease virus serotype O (FMDV-O), foot-and-mouth disease serotype A (FMDV-A), SVA-negative sera for determining cut-off value, SVA-positive serum used for sensitivity experiments, clinical samples for evaluating method compliance, along with the Senecavirus A strain used in VNT were preserved at the National Foot-and-Mouth Disease Reference Laboratory in Lanzhou, China.

**TABLE 2** Results of the repeatability assay for DAgS-ELISA

| Sample No. | | Intra-assay | | Inter-assay | |
|---|---|---|---|---|---|
| | | $\overline{X}$ ± SD | CV (%) | $\overline{X}$±SD | CV (%) |
| Positive samples | 1 | 1.057 ± 0.038 | 3.58 | 1.088 ± 0.038 | 3.53 |
| | 2 | 2.193 ± 0.075 | 3.44 | 2.251 ± 0.046 | 2.06 |
| | 3 | 2.035 ± 0.109 | 5.37 | 2.037 ± 0.039 | 1.90 |
| | 4 | 1.687 ± 0.030 | 1.77 | 1.688 ± 0.011 | 0.64 |
| | 5 | 1.860 ± 0.077 | 4.15 | 2.007 ± 0.048 | 2.39 |
| | 6 | 0.102 ± 0.005 | 4.78 | 0.095 ± 0.006 | 6.08 |
| Negative samples | 7 | 0.107 ± 0.007 | 7.00 | 0.109 ± 0.006 | 5.83 |
| | 8 | 0.112 ± 0.007 | 6.21 | 0.109 ± 0.009 | 8.55 |
| | 9 | 0.095 ± 0.005 | 5.11 | 0.086 ± 0.002 | 2.38 |
| | 10 | 0.102 ± 0.005 | 5.32 | 0.130 ± 0.008 | 5.89 |

**TABLE 3**  Comparison of DAgS-ELISA with VNT

| VNT | DAgS-ELISA | | Total |
|---|---|---|---|
| | **Positive** | **Negative** | |
| Positive | 19 | 0 | 19 |
| Negative | 9 | 138 | 147 |
| Total | 28 | 138 | 166 |
| Kappa value | | 0.78 | |

Ni Sepharose 6 Fast Flow (GE17-5318-01, Merck, Germany); goat anti-pig IgG HRP (Horseradish Peroxidase) antibody (ab6915, Abcam, Cambridge, UK); rabbit anti-mouse IgG HRP antibody (ab6728, Abcam, Cambridge, UK); thrombin (T8021, Solarbio Biotechnology Co., Ltd, Beijing, China); Ms mAb to 6 × His tag (ab202004, Abcam, Cambridge, UK); and HRP Conjugation Kit (ab102890, Abcam, Cambridge, UK).

## Expression and identification of the VP2 protein

The VP2 gene (GenBank ID: KY747510) was inserted into the pET-30a (+) vector, and the recombinant plasmid pET-30a-VP2 was transformed into BL21 (DE3) competent cells of *E. coli*. The single colony was expanded in Luria-Bertani medium containing kanamycin, and then, 0.1 mM isopropyl-β-D-thiogalatoside was added and maintained at 16°C for 18 h to further enhance the expression. After that, when the value of $OD_{600nm}$ reached 0.6, the bacteria were collected by centrifugation and then resuspended in phosphate-buffered saline (PBS) followed by sonicated twice, and finally, centrifuged at 12,000 rpm for 5 min at 4°C.

## Purification of VP2 protein and removal of his tag

The bacterial sediment was re-suspended and sonicated in a PBS buffer containing 20 mM Tris-HCl, 500 mM NaCl, and 1 mM EDTA at pH 8.0. The supernatant was centrifuged at 12,000 rpm for 5 min at 4°C, added to a nickel affinity chromatography column that was pre-equilibrated with an equilibration buffer (PBS with 50 mM Tris-HCl, 500 mM NaCl, and 3 mM EDTA at pH 8.5). Next, the column was washed with 40 mL of 50 mM imidazole, and the target protein was finally eluted with 500 mM imidazole.

After dialysis in 20 mM Tris-HCl, VP2 protein was mixed with thrombin at the ratio of 1 mg: 4 U and incubated at 37°C for 12 h, and then purified by nickel affinity chromatography. Subsequently, the VP2 protein without His-tag was obtained and then determined for concentration using the BCA protein assay kit (A53225, Thermo Fisher Scientific, USA) and stored at −80°C. In the next step, The HRP Conjugation Kit Lightning-Link (ab102890, Abcam, Shanghai, China) was utilized to conjugate the purified VP2 protein without His tag and HRP, according to the manufacturer's instructions; afterward, the HRP-labeled VP2 protein was used for the development and optimization of DAgS-ELISA. After expression, purification, and removal of His tag, the VP2 protein was analyzed and identified by sodium dodecyl sulfate-polyacrylamide gel electrophoresis (SDS-PAGE).

## Reactivity of VP2 protein

To visualize the specific reaction between VP2 protein and SVA serum, indirect ELISA and western blotting were performed. In the indirect ELISA, 50 µL diluted SVA serum (1:10) was first added to the plate pre-coated with VP2 protein (100 ng/well) and incubated at 37°C for 30 min. After four rounds of washing, HRP-labeled goat anti-pig IgG (1:30,000) was added and incubated at 37°C for 30 min. After the final washing and drying, 50 µL of 3, 3, 5, 5-tetramethylbenzidine (TMB) substrate solution was added and incubated at 37°C for 12 min. Afterward, 50 µL of 2 M sulfuric acid was added to each well to terminate the reaction, and then, the absorbance at $OD_{450nm}$ was recorded for each well using a microplate spectrophotometer. In the process of western blotting, the purified VP2 protein was first subjected to 12.5% SDS-PAGE electrophoresis, and then, the isolated

protein was transferred onto nitrocellulose (NC) membrane, which was blocked with 5% skim milk at room temperature (RT) for 2 h, washed with Tris-buffered saline containing 0.05% Tween 20 (TBST), and overnight incubated with SVA-positive and -negative serum (1:1,000) at 4℃. After washing thrice with TBST, HRP-labeled goat anti-pig IgG (1:10,000) was added under RT and incubated for 1 h. After the last step of washing, chemiluminescence analysis was performed.

Similarly, the specific reaction of VP2 protein with 6 × His tag antibody (6 × His tag mAb diluted at 1:5,000, HRP-labeled rabbit anti-mouse IgG diluted at 1:15,000) was identified by western blotting, and the remaining methods were as described above.

## Establishment and optimization of DAgS-ELISA method based on VP2 protein

In order to determine the optimal dilution ratio for our newly developed DAgS-ELISA, various dilution combinations of antigen and sera were used for checkerboard titration. Following this, the serum incubation time, dilution ratio, and incubation time for the HRP-labeled antigen, as well as TMB reaction time were all established. The detailed optimization process was as follows. The VP2 protein was diluted from 0.5 to 8.0 µg/mL with carbonate buffer solution (0.05M, pH 9.6) in a continuous 2-fold ratio. High-binding 96-well microplates (Costar 2592, Corning, USA) were incubated with the diluted VP2 protein solution 50 µL/well overnight at 4℃. After that, they were washed thrice with TBST and blocked with 250 µL of 5% skim milk in 0.01 mM TBST for 2 h at 37℃. After three washes with TBST, they were dried at RT. The standard sera were diluted at ratios ranging from 1:2 to 1:10 and then incubated for 15, 30, 45, and 60 min, respectively, to determine the optimal serum dilution ratio and incubation time. The VP2 protein conjugated with HRP was prepared according to the method described in 2.3 and diluted at the following ratios of 1:40,000; 1:60,000; 1:80,000; 1:100,000; and 1:120,000, with reaction time at 15, 30, 45, and 60 min, respectively, to screen and identify the best-combined conditions. After the final washing, 50 µL of TMB solution was added and incubated at 37℃ for 5, 8, 10, 12, 15, and 20 min, respectively, to select the optimal reaction time for TMB. Fifty microliters of 2 M sulfuric acid were added to terminate the colorimetric reaction, and the $OD_{450nm}$ was measured using a microplate spectrophotometer. All experiments were performed in triplicate.

## Cut-off value determination

The optimized DAgS-ELISA was conducted to evaluate 130 SVA-negative sera, and the $OD_{450nm}$ values of all samples were recorded. The cut-off value was defined as $\overline{X} + 3$ SD after calculating the average ($\overline{X}$) and standard deviation (SD). If the $OD_{450nm}$ value of the serum sample is higher than or equal to this cut-off value, it will be judged positive, and vice versa.

## Sensitivity and specificity evaluation

The SVA-positive serum samples were diluted in a continuous 2-fold ratio ranging from 1:8 to 1:1024 and tested under the optimized DAgS-ELISA conditions to assess sensitivity. Positive sera of SVA and common swine pathogens such as ASFV, CSFV, PRRSV, PRV, FMDV-A, and FMDV-O were detected under optimized conditions, and three serum samples were detected for each pathogen. The specificity of the method was evaluated by whether there was cross-reaction.

## Repeatability evaluation

Ten serum samples were used, including five negative and five positive, to evaluate the intra- and inter-assay repeatability. For the intra-assay, each serum sample was tested in five replicates on different plates, whereas for the inter-assay, they were tested five times on the same plate.

## Clinical samples detection

The established DAgS-ELISA and VNT were used simultaneously to detect 166 clinical serum samples from Gansu Province in 2021 to evaluate the clinical performance. The VNT experiment was independently repeated three times in the Biosafety Level II laboratory. In brief, after 30 minutes of thermal inactivation at 56°C, each serum was continuously diluted with DMEM medium at a 2-fold ratio (ranging from 1:4 to 1:512) into 96-well plates with 50 µL/well, and positive and negative controls were set for each plate. Subsequently, 50 µL SVA virus (100 TCID50 calculated by the Reed Muench method) was added to each well and incubated at 37°C for 1 h to neutralize the serum, and then, 100 µL of culture supernatant was successively transferred into the prepared BHK21 cells (106/mL), incubated at 37°C for 72 h, and the cytopathic effect (CPE) was observed the neutralization titer was calculated. Under the condition that both negative and positive quality control were established, if the titer of the tested serum is greater than or equal to 1:45, it is judged as SVA antibody positive; if it is lower than 1:16, considered negative; if the titer is between 1:16 and 1:45, it is suspicious and requires repeated testing; and if the retest titer is greater than or equal to 1:16, it is ultimately determined to be positive. By counting the number of positive and negative samples, the kappa value was calculated, and the results were then comprehensively analyzed.

## Statistics and analysis

The data were visually displayed through GraphPad Prism software (version 8.0.2; GraphPad Software, San Diego, CA, USA). Each sample was tested at least three times, and the results are presented in the form of average ($\overline{X}$) ± SD. The reproducibility was evaluated by the CV, and the degree of agreement between different methods was assessed using the kappa value.

### ACKNOWLEDGMENTS

The authors thank Home for Researchers editorial team (www.home-for-research-ers.com) for language editing service.

This work was supported by the Major Science and Technology Project of Gansu Province (23ZDNA007), the Innovation Consortium Project of Gansu Province (22ZD6NA012), the National Key R&D Program of China (2021YFD1801300 and 2021YFD1800100), the STI 2030-Major Projects (2023ZD0404301), Innovation Program of Chinese Academy of Agricultural Sciences (CAAS-CSLPDCP-202302 and CAAS-ASTIP-2024-LVRI), the Open Competition Program of Top Ten Critical Priorities of Agricultural Science and Technology Innovation for the 14th Five-Year Plan of Guangdong Province (2023SDZG02), China Agriculture Research System of Ministry of Finance and Ministry of Agriculture and Rural Affairs (CARS-35), Science and Technology Plan Project of Gansu Province (21IR7RA024), the Strategic Priority Research Program of Project of the National Center of Technology Innovation for Pigs (NCTIP-XD/C03).

### AUTHOR AFFILIATIONS

[1]State Key Laboratory for Animal Disease Control and Prevention, College of Veterinary Medicine, Lanzhou University, Lanzhou Veterinary Research Institute, Chinese Academy of Agricultural Sciences, Lanzhou, China
[2]Gansu Province Research Center for Basic Disciplines of Pathogen Biology, Lanzhou, China

### AUTHOR ORCIDs

Jie Chen  http://orcid.org/0009-0007-1548-5214
Hong Tian  http://orcid.org/0000-0002-1764-4916
Haixue Zheng  http://orcid.org/0000-0002-1764-4916

## FUNDING

| Funder | Grant(s) | Author(s) |
|---|---|---|
| the major science and technology project of gansu province | 23ZDNA007 | Haixue Zheng |
| the innovation consortium project of gansu province | 22ZD6NA012 | Haixue Zheng |
| MOST \| National Key Research and Development Program of China (NKPs) | 2021YFD1801300, 2021YFD1800100 | Haixue Zheng |
| the STI 2030 major projects | 2023ZD0404301 | Haixue Zheng |
| innovation program of chinese academy of agricultural sciences | CAAS-CSLPDCP-202302, CAAS-ASTIP-2024-LVRI | Haixue Zheng |
| the open competition program of top ten critical priorities of agricultural science and technology innovation for the 14th five-year plan of guangdong province | 2023SDZG02 | Haixue Zheng |
| China Agriculture Research System of Ministry of Finance and Ministry of Agriculture and Rural Affairs | CARS-35 | Haixue Zheng |
| Science and Technology Plan Project of Gansu Province | 21IR7RA024 | Haixue Zheng |
| the Strategic Priority Research Program of Project of the National Center of Technology Innovation for Pigs | NCTIP-XD/C03 | Haixue Zheng |

## AUTHOR CONTRIBUTIONS

Jie Chen, Data curation, Software, Validation, Visualization, Writing – original draft, Writing – review and editing | Zhengwang Shi, Methodology | Juncong Luo, Methodology | Caixia Jia, Investigation | Xiaoyang Zhang, Resources | Juanjuan Wei, Resources | Shuaipeng Li, Resources | Yuqian Zhu, Software | Tao Xi, Software | Jing Zhou, Software | Yindi He, Supervision | Xintai Shi, Investigation | Huanchen Liao, Resources | Hong Tian, Data curation, Formal analysis, Project administration | Haixue Zheng, Data curation, Formal analysis, Project administration

## DATA AVAILABILITY

The sequence of VP2 protein is available from the corresponding author upon reasonable request.

## ADDITIONAL FILES

The following material is available online.

Open Peer Review

**PEER REVIEW HISTORY (review-history.pdf).** An accounting of the reviewer comments and feedback.

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
