## [Reviewer comments · Microbiology Spectrum]

Microbiology Spectrum

Development and Optimization of a Double Antigen Sandwich ELISA Detecting for Senecavirus A Antibodies Based on VP2 Protein

Jie Chen, Zhengwang Shi, Juncong Luo, Caixia Jia, Xiaoyang Zhang, Juanjuan Wei, Shuaipeng Li, Yuqian Zhu, Tao Xi, Jing Zhou, Yindi He, Xintai Shi, Huancheng Liao, Hong Tian, and Haixue Zheng

Corresponding Author(s): Jie Chen, Lanzhou Veterinary Research Institute, Chinese Academy of Agricultural Sciences

Review Timeline:

Submission Date:	August 16, 2024
Editorial Decision:	September 8, 2024
Revision Received:	September 13, 2024
Editorial Decision:	September 13, 2024
Revision Received:	September 14, 2024
Accepted:	September 16, 2024

Editor: Benjamin Liu

Reviewer(s): Disclosure of reviewer identity is with reference to reviewer comments included in decision letter(s). The following individuals involved in review of your submission have agreed to reveal their identity: Kaan Çeylan (Reviewer #2)

Transaction Report:

DOI: <https://doi.org/10.1128/spectrum.02043-24>

Re: Spectrum02043-24 (Development and Optimization of a Double Antigen Sandwich ELISA Detecting for Senecavirus A Antibodies Based on VP2 Protein)

Dear Dr. Jie Chen:

Thank you for the privilege of reviewing your work. Below you will find my comments, instructions from the Spectrum editorial office, and the reviewer comments.

Editor's comments:

Line 91-96: The authors should improve the introduction on pros and cons of ELISA for the detection of antibodies in comparison with PCR. Please add the following in the Introduction: PCR's major limitation is the sensitivity of PCR depends upon the timing of appropriate sample collection, though it is very sensitive and specific. In contrast, the limitation of serology is window period and cross-reaction issue, though they are fast and inexpensive. More references should be cited, with the following ones as examples (citing is optional):

Epidemiological and clinical overview of the 2024 Oropouche virus disease outbreaks, an emerging/re-emerging neurotropic arboviral disease and global public health threat. *J Med Virol.* 2024 Sep;96(9):e29897. doi: 10.1002/jmv.29897. PMID: 39221481.

Laboratory diagnosis of CNS infections in children due to emerging and re-emerging neurotropic viruses. *Pediatr Res.* 2024 Jan;95(2):543-550. doi: 10.1038/s41390-023-02930-6. Epub 2023 Dec 2. PMID: 38042947.

Mpox (Monkeypox) Virus and Its Co-Infection with HIV, Sexually Transmitted Infections, or Bacterial Superinfections: Double Whammy or a New Prime Culprit? *Viruses.* 2024 May 15;16(5):784. doi: 10.3390/v16050784. PMID: 38793665; PMCID: PMC11125633.

Novel HBV recombinants between genotypes B and C in 3'-terminal reverse transcriptase (RT) sequences are associated with enhanced viral DNA load, higher RT point mutation rates and place of birth among Chinese patients. *Infect Genet Evol.* 2018 Jan;57:26-35. doi: 10.1016/j.meegid.2017.10.023. Epub 2017 Oct 27. PMID: 29111272.

Naturally occurring deletions/insertions in HBV core promoter tend to decrease in hepatitis B e antigen-positive chronic hepatitis B patients during antiviral therapy. *Antivir Ther.* 2015;20(6):623-32. doi: 10.3851/IMP2955. Epub 2015 Apr 2. PMID: 25838313.

Please return the manuscript within 30 days; if you cannot complete the modification within this time period, please contact me. If you do not wish to modify the manuscript and prefer to submit it to another journal, notify me immediately so that the manuscript may be formally withdrawn from consideration by Spectrum.

Revision Guidelines

Data availability: ASM policy requires that data be available to the public upon online posting of the article, so please verify all links to sequence records, if present, and make sure that each number retrieves the full record of the data. If a new accession

number is not linked or a link is broken, provide Spectrum production staff with the correct URL for the record. If the accession numbers for new data are not publicly accessible before the expected online posting of the article, publication may be delayed; please contact production staff (Spectrum@asmusa.org) immediately with the expected release date.

Sincerely,
Benjamin Liu
Editor
Microbiology Spectrum

Reviewer #1 (Comments for the Author):

The development of diagnostic assays for emerging viral pathogens, such as Senecavirus A (SVA), is critical for effective disease management and control. This manuscript presents the development of an ELISA-based diagnostic test, which is particularly relevant given the journal's focus on advancing diagnostic methodologies. The test's uniqueness lies in its application to SVA, a pathogen for which such a diagnostic approach has not previously been explored, making this study a valuable and novel contribution to the field. Given the clinical and economic importance of accurately distinguishing SVA from other similar diseases, the proposed approach is both timely and relevant. But here, I have some suggestions to enhance the clarity and impact of the manuscript, which I believe could further strengthen its contribution to the field.

- In the manuscript, please ensure that all materials are cited according to the standard format. Specifically, right after the material name, include catalogue number (if necessary), brand, city, and country. For instance: 'DNA Isolation Kit (Catalogue Number XYZ123, BioTech Solutions, Berlin, Germany)'. Adhering to this format will enhance clarity and consistency in your references.
- In line 472: In Fig. 2A there is no need for additional statement under the graph. Please remove the nonessential legend "reaction between VP2 protein and SVA serum"
- For all graphs presented in Fig. 3, statistical analyses should be conducted specific to the parameters being evaluated between groups. For instance, it is necessary to determine whether the differences between data obtained from serum incubation samples at 15 and 30 minutes are statistically significant. Therefore, each graph should include statistical analysis of the positive serum samples against each relevant parameter to address such questions.
- In Fig. 5A, it appears that only one serum per group was used for cross-reactivity analysis. For more robust results, it is recommended to use at least three samples per group for cross-reactivity studies. Utilizing multiple samples per group will improve the statistical power and reliability of the cross-reactivity analysis
- In line 137: Please quantify the sensitivity and specificity of the developed assay in percentage terms. Providing sensitivity and specificity as percentages is crucial because it offers a clear and precise measure of the assay's performance. Percentages allow for straightforward comparison with other assays and enable a better understanding of the assay's effectiveness in detecting true positives and negatives, which is essential for evaluating its diagnostic or analytical accuracy.
- In line 148, you mention that only 9 of the same serum samples were positive with DAgS-ELISA but negative with VNT. While this is understandable, a brief commentary on the underlying reasons for this discrepancy could be valuable. As scientific articles are among the best educational tools for younger scientists, providing such insights is important. Additionally, this presents an excellent opportunity to highlight the advantages of your developed ELISA test over the VNT test, emphasizing its potential superiority.
- Also in line 30, you mention a kappa value of 0.78 to highlight the good agreement with the gold-standard VNT test. While this is an important point, it would also be valuable to emphasize that your developed test captures not only neutralizing antibodies but all SVA-specific antibodies, which results in higher sensitivity. Highlighting this aspect would further underscore the significance of your work.
- In line 284, additional information regarding who conducted the VNT test, where it was performed, and the methodology used would be beneficial. There is no information provided regarding the biosafety measures taken for VNT test. Providing these details will enhance the transparency and reproducibility of the study.
- Also, it is unclear whether the serum samples used were inactivated. Providing these details is essential to ensure that appropriate biosafety protocols were followed.

Reviewer #2 (Comments for the Author):

Dear author,

The review of the study titled Development and Optimization of a Double Antigen Sandwich ELISA Detecting for Senecavirus A Antibodies Based on VP2 Protein has been completed. I recommend that you pay attention to the following points.

1. The introduction section has no title.
2. The information in the sentence between lines 89 and 91 must be referenced.
3. Information should be briefly provided on the clinical course of the disease caused by Senecavirus A in pigs and why it is important for humans.

I wish good work

Comments to authors:

The development of diagnostic assays for emerging viral pathogens, such as Senecavirus A (SVA), is critical for effective disease management and control. This manuscript presents the development of an ELISA-based diagnostic test, which is particularly relevant given the journal's focus on advancing diagnostic methodologies. The test's uniqueness lies in its application to SVA, a pathogen for which such a diagnostic approach has not previously been explored, making this study a valuable and novel contribution to the field. Given the clinical and economic importance of accurately distinguishing SVA from other similar diseases, the proposed approach is both timely and relevant. But here, I have some suggestions to enhance the clarity and impact of the manuscript, which I believe could further strengthen its contribution to the field.

- In the manuscript, please ensure that all materials are cited according to the standard format. Specifically, right after the material name, include catalogue number (if necessary), brand, city, and country. For instance: 'DNA Isolation Kit (Catalogue Number XYZ123, BioTech Solutions, Berlin, Germany)'. Adhering to this format will enhance clarity and consistency in your references.
- In line 472: In Fig. 2A there is no need for additional statement under the graph. Please remove the nonessential legend "reaction between VP2 protein and SVA serum"
- For all graphs presented in Fig. 3, statistical analyses should be conducted specific to the parameters being evaluated between groups. For instance, it is necessary to determine whether the differences between data obtained from serum incubation samples at 15 and 30 minutes are statistically significant. Therefore, each graph should include statistical analysis of the positive serum samples against each relevant parameter to address such questions.
- In Fig. 5A, it appears that only one serum per group was used for cross-reactivity analysis. For more robust results, it is recommended to use at least three samples per group for cross-reactivity studies. Utilizing multiple samples per group will improve the statistical power and reliability of the cross-reactivity analysis.
- In line 137: Please quantify the sensitivity and specificity of the developed assay in percentage terms. Providing sensitivity and specificity as percentages is crucial because it offers a clear and precise measure of the assay's performance. Percentages allow for straightforward comparison with other assays and enable a better

understanding of the assay's effectiveness in detecting true positives and negatives, which is essential for evaluating its diagnostic or analytical accuracy.

- In line 148, you mention that only 9 of the same serum samples were positive with DAgS-ELISA but negative with VNT. While this is understandable, a brief commentary on the underlying reasons for this discrepancy could be valuable. As scientific articles are among the best educational tools for younger scientists, providing such insights is important. Additionally, this presents an excellent opportunity to highlight the advantages of your developed ELISA test over the VNT test, emphasizing its potential superiority.
- Also in line 30, you mention a kappa value of 0.78 to highlight the good agreement with the gold-standard VNT test. While this is an important point, it would also be valuable to emphasize that your developed test captures not only neutralizing antibodies but all SVA-specific antibodies, which results in higher sensitivity. Highlighting this aspect would further underscore the significance of your work.
- In line 284, additional information regarding who conducted the VNT test, where it was performed, and the methodology used would be beneficial. There is no information provided regarding the biosafety measures taken for VNT test. Providing these details will enhance the transparency and reproducibility of the study.
- Also, it is unclear whether the serum samples used were inactivated. Providing these details is essential to ensure that appropriate biosafety protocols were followed.

Comments to editor:

The study aims to develop a novel diagnostic assay for Senecavirus A (SVA) using a double antigen sandwich ELISA approach. This is a crucial area of research given the potential impact of SVA in various contexts. The manuscript provides a detailed description of the assay development process, including the optimization of experimental conditions and validation of the test's performance. The study's approach is good enough and the results show potential for significant impact. It is written in clear and accessible language. While this approach facilitates understanding, the manuscript appears to have some shortcomings in terms of scientific depth and intellectual rigor. There is room for improvement in presenting a more thorough analysis and discussion of the results.

During the review process, I encountered some uncertainty regarding the biosafety concerns mentioned in the manuscript. While Senecavirus A is generally classified as a BSL-2 pathogen, the recommendation for serum inactivation in ELISA assays appears more as a precaution rather than a strict requirement. The reviewed manuscript does not provide any information on serum inactivation. Additionally, there is no information on how the VNT test was conducted or whether specific biosafety measures were taken, and the literature does not fully clarify these aspects. I wish to express my uncertainty regarding these issues.

While the statistical analyses reported in the study are correct, they appear to be incomplete. For the graphs presented in Fig. 3, I suggest conducting additional statistical analyses specific to the parameters being evaluated. To better evaluate the impact of the parameters optimized during the experiments, it would be useful to determine whether the differences of the tested samples are statistically significant. Additional details or further statistical evaluations could enhance the robustness and interpretability of the results.

Dear Editors,

Thank you for your letter concerning our manuscript entitled “Development and Optimization of a Double Antigen Sandwich ELISA Detecting for Senecavirus A Antibodies Based on VP2 Protein” (**manuscript ID: Spectrum02043-24**). Those comments are valuable and very helpful for revising and improving our manuscript.

We have carefully taken those comments into consideration and have made corrections. Revised portions are marked using ‘track changes’ in the revised manuscript. The marked revision file and unmarked revision file, as well as the supplementary material containing a supplementary figure are attached. And below are our detailed responses to their comments.

We look forward to working with you and the reviewers to make this manuscript closer to publication in Microbiology Spectrum. If you have any other questions, please contact us without hesitation. We look forward to hearing from you.

Sincerely yours,

Jie Chen

Point-by-Point Response to reviewers’ comments

Editor

Line 91-96: The authors should improve the introduction on pros and cons of ELISA for the detection of antibodies in comparison with PCR. Please add the following in the Introduction: PCR's major limitation is the sensitivity of PCR depends upon the timing of appropriate sample collection, though it is very sensitive and specific. In contrast, the limitation of serology is window period and cross-reaction issue, though they are fast and inexpensive. More references should be cited, with the following ones as examples (citing is optional).

Response : Thank you for your valuable comments. We appreciated and supplemented the relevant description in line 94-103 and added references.

Reviewer#1

- In the manuscript, please ensure that all materials are cited according to the standard format.

Specifically, right after the material name, include catalogue number (if necessary), brand, city, and country. For instance: 'DNA Isolation Kit (Catalogue Number XYZ123, BioTech Solutions, Berlin, Germany)'. Adhering to this format will enhance clarity and consistency in your references.

Response: Thank you for your kind advice. The relevant additions and modifications have been made in line 221-227.

- In line 472: In Fig. 2A there is no need for additional statement under the graph. Please remove the nonessential legend "reaction between VP2 protein and SVA serum".

Response: Thank you for your kind reminding us. The legend of Fig. 2A has been removed as shown below.

FIG 2 Reaction of purified VP2 protein
(Notes: The legend was removed)

- For all graphs presented in Fig. 3, statistical analyses should be conducted specific to the parameters being evaluated between groups. For instance, it is necessary to determine whether the differences between data obtained from serum incubation samples at 15 and 30 minutes are statistically significant. Therefore, each graph should include statistical analysis of the positive serum samples against each relevant parameter to address such questions.

Response: We appreciate your valuable comments. All the P/N value (**shown as line charts rather than bar charts**) in Fig. 3 were analyzed statistically through multiple comparisons during one-way ANOVA, basically, there was no statistical significance among the groups under other conditions except for serum incubation for 15 and 60 min, so we only made supplementary markers in Fig. 3A. The difference between other groups was not statistically significant, so no additional modifications were made, but this did not affect the screening and optimization of DAgs-ELISA conditions, because the maximum P/N value was taken as the selection criterion, which also complied with the principle of checkerboard titration.

The following are the revised Fig. 3 and the detailed data for multiple comparisons in the one-way ANOVA analysis under different conditions which labeled 'Supplementary FIG A to D', in order to be distinguished from the figures in the manuscript.

FIG 3 Optimization results for the DAgS-ELISA
(Notes: Only FIG 3A was modified with statistical analysis)

Ordinary one-way ANOVA								
Multiple comparisons								
Number of families	1							
Number of comparisons per family	6							
Alpha	0.05							
Tukey's multiple comparisons test	Mean Diff.	95.00% CI of diff.	Significant?	Summary	Adjusted P Value			
Serum-15min vs. Serum-30min	-1.925	-4.472 to 0.6216	No	ns	0.1500	A-B		
Serum-15min vs. Serum-45min	-1.822	-4.368 to 0.7251	No	ns	0.1794	A-C		
Serum-15min vs. Serum-60min	-2.675	-5.222 to -0.1280	Yes	*	0.0399	A-D		
Serum-30min vs. Serum-45min	0.1035	-2.443 to 2.650	No	ns	0.9991	B-C		
Serum-30min vs. Serum-60min	-0.7496	-3.296 to 1.797	No	ns	0.7839	B-D		
Serum-45min vs. Serum-60min	-0.8531	-3.400 to 1.694	No	ns	0.7145	C-D		
Test details	Mean 1	Mean 2	Mean Diff.	SE of diff.	n1	n2	q	DF
Serum-15min vs. Serum-30min	11.93	13.85	-1.925	0.7953	3	3	3.423	8
Serum-15min vs. Serum-45min	11.93	13.75	-1.822	0.7953	3	3	3.239	8
Serum-15min vs. Serum-60min	11.93	14.60	-2.675	0.7953	3	3	4.757	8
Serum-30min vs. Serum-45min	13.85	13.75	0.1035	0.7953	3	3	0.1840	8
Serum-30min vs. Serum-60min	13.85	14.60	-0.7496	0.7953	3	3	1.333	8
Serum-45min vs. Serum-60min	13.75	14.60	-0.8531	0.7953	3	3	1.517	8

Supplementary FIG A
One-way ANOVA analysis in reaction time of serum

Ordinary one-way ANOVA Multiple comparisons								
Number of families	1							
Number of comparisons per family	10							
Alpha	0.05							
Tukey's multiple comparisons test	Mean Diff.	95.00% CI of diff.	Significant?	Summary	Adjusted P Value			
1:40000 vs. 1:60000	-1.006	-7.384 to 5.372	No	ns	0.9833	A-B		
1:40000 vs. 1:80000	-1.775	-8.153 to 4.603	No	ns	0.8845	A-C		
1:40000 vs. 1:100000	0.7939	-5.584 to 7.172	No	ns	0.9931	A-D		
1:40000 vs. 1:120000	0.8798	-5.498 to 7.258	No	ns	0.9899	A-E		
1:60000 vs. 1:80000	-0.7683	-7.146 to 5.610	No	ns	0.9939	B-C		
1:60000 vs. 1:100000	1.800	-4.578 to 8.178	No	ns	0.8794	B-D		
1:60000 vs. 1:120000	1.886	-4.492 to 8.264	No	ns	0.8612	B-E		
1:80000 vs. 1:100000	2.569	-3.809 to 8.946	No	ns	0.6831	C-D		
1:80000 vs. 1:120000	2.654	-3.724 to 9.032	No	ns	0.6582	C-E		
1:100000 vs. 1:120000	0.08588	-6.292 to 6.464	No	ns	>0.9999	D-E		
Test details	Mean 1	Mean 2	Mean Diff.	SE of diff.	n1	n2	q	DF
1:40000 vs. 1:60000	11.23	12.24	-1.006	1.938	3	3	0.7343	10
1:40000 vs. 1:80000	11.23	13.01	-1.775	1.938	3	3	1.295	10
1:40000 vs. 1:100000	11.23	10.44	0.7939	1.938	3	3	0.5794	10
1:40000 vs. 1:120000	11.23	10.35	0.8798	1.938	3	3	0.6420	10
1:60000 vs. 1:80000	12.24	13.01	-0.7683	1.938	3	3	0.5607	10
1:60000 vs. 1:100000	12.24	10.44	1.800	1.938	3	3	1.314	10
1:60000 vs. 1:120000	12.24	10.35	1.886	1.938	3	3	1.376	10
1:80000 vs. 1:100000	13.01	10.44	2.569	1.938	3	3	1.874	10
1:80000 vs. 1:120000	13.01	10.35	2.654	1.938	3	3	1.937	10
1:100000 vs. 1:120000	10.44	10.35	0.08588	1.938	3	3	0.06267	10

Supplementary FIG B
One-way ANOVA analysis in dilution ratio of HRP-labeled antigen

Ordinary one-way ANOVA Multiple comparisons								
Number of families	1							
Number of comparisons per family	6							
Alpha	0.05							
Tukey's multiple comparisons test	Mean Diff.	95.00% CI of diff.	Significant?	Summary	Adjusted P Value			
15min vs. 30min	-0.3065	-5.119 to 4.506	No	ns	0.9967	A-B		
15min vs. 45min	-0.3268	-5.139 to 4.486	No	ns	0.9961	A-C		
15min vs. 60min	3.740	-1.072 to 8.553	No	ns	0.1365	A-D		
30min vs. 45min	-0.02033	-4.833 to 4.792	No	ns	>0.9999	B-C		
30min vs. 60min	4.047	-0.7657 to 8.859	No	ns	0.1026	B-D		
45min vs. 60min	4.067	-0.7453 to 8.879	No	ns	0.1007	C-D		
Test details	Mean 1	Mean 2	Mean Diff.	SE of diff.	n1	n2	q	DF
15min vs. 30min	14.80	15.11	-0.3065	1.503	3	3	0.2884	8
15min vs. 45min	14.80	15.13	-0.3268	1.503	3	3	0.3076	8
15min vs. 60min	14.80	11.06	3.740	1.503	3	3	3.520	8
30min vs. 45min	15.11	15.13	-0.02033	1.503	3	3	0.01914	8
30min vs. 60min	15.11	11.06	4.047	1.503	3	3	3.808	8
45min vs. 60min	15.13	11.06	4.067	1.503	3	3	3.827	8

Supplementary FIG C
One-way ANOVA analysis in reaction time for HRP-labeled antigen

Ordinary one-way ANOVA								
Multiple comparisons								
Number of families	1							
Number of comparisons per family	15							
Alpha	0.05							
Tukey's multiple comparisons test	Mean Diff.	95.00% CI of diff.	Significant?	Summary	Adjusted P Value			
TMB-5min vs. TMB-8min	-1.290	-5.430 to 2.850	No	ns	0.8930			A-B
TMB-5min vs. TMB-10min	-0.3704	-4.511 to 3.770	No	ns	0.9996			A-C
TMB-5min vs. TMB-12min	-0.3065	-4.447 to 3.834	No	ns	0.9998			A-D
TMB-5min vs. TMB-15min	-3.650	-7.791 to 0.4900	No	ns	0.0964			A-E
TMB-5min vs. TMB-20min	-3.520	-7.660 to 0.6206	No	ns	0.1144			A-F
TMB-8min vs. TMB-10min	0.9194	-3.221 to 5.060	No	ns	0.9717			B-C
TMB-8min vs. TMB-12min	0.9833	-3.157 to 5.124	No	ns	0.9625			B-D
TMB-8min vs. TMB-15min	-2.361	-6.501 to 1.780	No	ns	0.4385			B-E
TMB-8min vs. TMB-20min	-2.230	-6.370 to 1.910	No	ns	0.4951			B-F
TMB-10min vs. TMB-12min	0.06394	-4.076 to 4.204	No	ns	>0.9999			C-D
TMB-10min vs. TMB-15min	-3.280	-7.420 to 0.8604	No	ns	0.1554			C-E
TMB-10min vs. TMB-20min	-3.149	-7.290 to 0.9910	No	ns	0.1827			C-F
TMB-12min vs. TMB-15min	-3.344	-7.484 to 0.7965	No	ns	0.1433			D-E
TMB-12min vs. TMB-20min	-3.213	-7.354 to 0.9271	No	ns	0.1689			D-F
TMB-15min vs. TMB-20min	0.1306	-4.010 to 4.271	No	ns	>0.9999			E-F
Test details	Mean 1	Mean 2	Mean Diff.	SE of diff.	n1	n2	q	DF
TMB-5min vs. TMB-8min	11.67	12.96	-1.290	1.233	3	3	1.480	12
TMB-5min vs. TMB-10min	11.67	12.04	-0.3704	1.233	3	3	0.4250	12
TMB-5min vs. TMB-12min	11.67	11.98	-0.3065	1.233	3	3	0.3517	12
TMB-5min vs. TMB-15min	11.67	15.32	-3.650	1.233	3	3	4.188	12
TMB-5min vs. TMB-20min	11.67	15.19	-3.520	1.233	3	3	4.038	12
TMB-8min vs. TMB-10min	12.96	12.04	0.9194	1.233	3	3	1.055	12
TMB-8min vs. TMB-12min	12.96	11.98	0.9833	1.233	3	3	1.128	12
TMB-8min vs. TMB-15min	12.96	15.32	-2.361	1.233	3	3	2.708	12
TMB-8min vs. TMB-20min	12.96	15.19	-2.230	1.233	3	3	2.558	12
TMB-10min vs. TMB-12min	12.04	11.98	0.06394	1.233	3	3	0.07335	12
TMB-10min vs. TMB-15min	12.04	15.32	-3.280	1.233	3	3	3.763	12
TMB-10min vs. TMB-20min	12.04	15.19	-3.149	1.233	3	3	3.613	12
TMB-12min vs. TMB-15min	11.98	15.32	-3.344	1.233	3	3	3.836	12
TMB-12min vs. TMB-20min	11.98	15.19	-3.213	1.233	3	3	3.687	12
TMB-15min vs. TMB-20min	15.32	15.19	0.1306	1.233	3	3	0.1498	12

Supplementary FIG D
One-way ANOVA analysis in reaction time of TMB

- In Fig. 5A, it appears that only one serum per group was used for cross-reactivity analysis. For more robust results, it is recommended to use at least three samples per group for cross-reactivity studies. Utilizing multiple samples per group will improve the statistical power and reliability of the cross-reactivity analysis.

Response: We value your precious advice. In our previous specificity experiment, only one serum for each pathogen was used for cross-reactivity analysis in each group, which was indeed not rigorous and reliable. According to your suggestion, we re-conducted the cross-reactivity experiment, and three samples were used in each group for detection and verification, Fig. 5A has been corrected, and the corresponding description has also been updated and modified in line 140 and line 305.

FIG 5A Specificity of the DA_gS-ELISA.

(Notes: Data were visually displayed after re-conducted the cross-reactivity analysis with three samples)

- In line 137: Please quantify the sensitivity and specificity of the developed assay in percentage terms. Providing sensitivity and specificity as percentages is crucial because it offers a clear and precise measure of the assay's performance. Percentages allow for straightforward comparison with other assays and enable a better understanding of the assay's effectiveness in detecting true positives and negatives, which is essential for evaluating its diagnostic or analytical accuracy.

Response: Thank you for your valuable suggestion. The specificity analysis performed in our study was to simultaneously detect clinically positive serum samples of common swine pathogens, among them, FMDV was of highly similar symptoms to that of SVA infection. The OD₄₅₀ value was compared with the cut-off value to determine whether it was negative or positive, and the specificity was further evaluated by whether it was cross-reactive with other pathogens. Additionally, the sensitivity analysis involved in this study was to dilute positive serum samples by a continuous two-fold ratio ranging from 1:8 to 1:1024, which was finally evaluated by the limit of detection.

Although it is indeed more accurate and intuitive to present the data in the form of percentage quantification, I am afraid it is not suitable for the above evaluation method adopted in this study. In general, this study conforms to the evaluation principles of specificity and sensitivity, and does not affect the judgment of the final results.

FIG 5 Specificity and sensitivity of the DAgS-ELISA.

- In line 148, you mention that only 9 of the same serum samples were positive with DAgS-ELISA but negative with VNT. While this is understandable, a brief commentary on the underlying reasons for this discrepancy could be valuable. As scientific articles are among the best educational tools for younger scientists, providing such insights is important. Additionally, this presents an excellent opportunity to highlight the advantages of your developed ELISA test over the VNT test, emphasizing its potential superiority.

Response: Thank you for your valuable opinions Related supplements and explanations are made in the discussion section line 180-188.

- Also in line 30, you mention a kappa value of 0.78 to highlight the good agreement with the gold-standard VNT test. While this is an important point, it would also be valuable to emphasize that your developed test captures not only neutralizing antibodies but all SVA-specific antibodies, which results in higher sensitivity. Highlighting this aspect would further underscore the significance of your work.

Response: Thank you for your kind reminding. We have explored this in depth in the discussion section line 180-188.

- In line 284, additional information regarding who conducted the VNT test, where it was performed, and the methodology used would be beneficial. There is no information provided regarding the biosafety measures taken for VNT test. Providing these details will enhance the transparency and reproducibility of the study.

Response: Detailed procedures for VNT test are added in line 311-324.

- Also, it is unclear whether the serum samples used were inactivated. Providing these details is essential to ensure that appropriate biosafety protocols were followed.

Response: The serum samples used in the VNT test have been inactivated at 56°C for 30 min to ensure biosafety, and the detailed process is described in line 311-324.

Reviewer #2:

1. The introduction section has no title.

Response : Thank you for your kind reminding, the title of the introduction has been added to the line 50.

2. The information in the sentence between lines 89 and 91 must be referenced.

Response : The reference has been added and numbered.

3. Information should be briefly provided on the clinical course of the disease caused by Senecavirus A in pigs and why it is important for humans.

Response : Thank you for your valuable advice. The relative descriptions were mentioned in line 79-83.

Other changes

Besides the above correction, we tried our best to improve the manuscript and made some minor changes in the manuscript, which didn't list here but marked using 'track changes' in the marked manuscript. Beside the changes above, we have carefully checked the manuscript and not found any other questions.

We gratefully appreciate for Editors valuable comments and suggestions on our manuscript. We hope that the correction will meet with approval.

Re: Spectrum02043-24R1 (Development and Optimization of a Double Antigen Sandwich ELISA Detecting for Senecavirus A Antibodies Based on VP2 Protein)

Dear Dr. Jie Chen:

Thank you for the privilege of reviewing your work. Below you will find my comments, instructions from the Spectrum editorial office, and the reviewer comments.

Editor comments:

The revised version has addressed most of the questions.
Ref38 and Ref39 should have page number and correct doi information.
Ref38 should contain: "Viruses.16:784. doi: 10.3390/v16050784."
Ref39 should contain: "J Med Virol. 96:e29897. doi: 10.1002/jmv.29897."

Please return the manuscript within 7 days; if you cannot complete the modification within this time period, please contact me. If you do not wish to modify the manuscript and prefer to submit it to another journal, notify me immediately so that the manuscript may be formally withdrawn from consideration by Spectrum.

Revision Guidelines

Sincerely,
Benjamin Liu
Editor
Microbiology Spectrum

Dear Editors,

Thank you for your letter concerning our manuscript entitled “Development and Optimization of a Double Antigen Sandwich ELISA Detecting for Senecavirus A Antibodies Based on VP2 Protein” (**manuscript ID: Spectrum02043-24**). Those comments are valuable and very helpful for revising and improving our manuscript.

We have carefully taken those comments into consideration and have made corrections. Revised portions are marked using ‘track changes’ in the revised manuscript. The marked revision file and unmarked revision file, as well as the supplementary material containing a supplementary figure are attached. And below are our detailed responses to their comments.

We look forward to working with you and the reviewers to make this manuscript closer to publication in Microbiology Spectrum. If you have any other questions, please contact us without hesitation. We look forward to hearing from you.

Sincerely yours,

Jie Chen

Point-by-Point Response to reviewers’ comments

Editor

Line 91-96: The authors should improve the introduction on pros and cons of ELISA for the detection of antibodies in comparison with PCR. Please add the following in the Introduction: PCR's major limitation is the sensitivity of PCR depends upon the timing of appropriate sample collection, though it is very sensitive and specific. In contrast, the limitation of serology is window period and cross-reaction issue, though they are fast and inexpensive. More references should be cited, with the following ones as examples (citing is optional).

Response : Thank you for your valuable comments. We appreciated and supplemented the relevant description in line 94-103 and added references.

Reviewer#1

- In the manuscript, please ensure that all materials are cited according to the standard format.

Specifically, right after the material name, include catalogue number (if necessary), brand, city, and country. For instance: 'DNA Isolation Kit (Catalogue Number XYZ123, BioTech Solutions, Berlin, Germany)'. Adhering to this format will enhance clarity and consistency in your references.

Response: Thank you for your kind advice. The relevant additions and modifications have been made in line 221-227.

- In line 472: In Fig. 2A there is no need for additional statement under the graph. Please remove the nonessential legend "reaction between VP2 protein and SVA serum".

Response: Thank you for your kind reminding us. The legend of Fig. 2A has been removed as shown below.

FIG 2 Reaction of purified VP2 protein
(Notes: The legend was removed)

- For all graphs presented in Fig. 3, statistical analyses should be conducted specific to the parameters being evaluated between groups. For instance, it is necessary to determine whether the differences between data obtained from serum incubation samples at 15 and 30 minutes are statistically significant. Therefore, each graph should include statistical analysis of the positive serum samples against each relevant parameter to address such questions.

Response: We appreciate your valuable comments. All the P/N value (**shown as line charts rather than bar charts**) in Fig. 3 were analyzed statistically through multiple comparisons during one-way ANOVA, basically, there was no statistical significance among the groups under other conditions except for serum incubation for 15 and 60 min, so we only made supplementary markers in Fig. 3A. The difference between other groups was not statistically significant, so no additional modifications were made, but this did not affect the screening and optimization of DAgs-ELISA conditions, because the maximum P/N value was taken as the selection criterion, which also complied with the principle of checkerboard titration.

The following are the revised Fig. 3 and the detailed data for multiple comparisons in the one-way ANOVA analysis under different conditions which labeled 'Supplementary FIG A to D', in order to be distinguished from the figures in the manuscript.

FIG 3 Optimization results for the DAGS-ELISA
(Notes: Only FIG 3A was modified with statistical analysis)

Ordinary one-way ANOVA								
Multiple comparisons								
Number of families	1							
Number of comparisons per family	6							
Alpha	0.05							
Tukey's multiple comparisons test	Mean Diff.	95.00% CI of diff.	Significant?	Summary	Adjusted P Value			
Serum-15min vs. Serum-30min	-1.925	-4.472 to 0.6216	No	ns	0.1500	A-B		
Serum-15min vs. Serum-45min	-1.822	-4.368 to 0.7251	No	ns	0.1794	A-C		
Serum-15min vs. Serum-60min	-2.675	-5.222 to -0.1280	Yes	*	0.0399	A-D		
Serum-30min vs. Serum-45min	0.1035	-2.443 to 2.650	No	ns	0.9991	B-C		
Serum-30min vs. Serum-60min	-0.7496	-3.296 to 1.797	No	ns	0.7839	B-D		
Serum-45min vs. Serum-60min	-0.8531	-3.400 to 1.694	No	ns	0.7145	C-D		
Test details	Mean 1	Mean 2	Mean Diff.	SE of diff.	n1	n2	q	DF
Serum-15min vs. Serum-30min	11.93	13.85	-1.925	0.7953	3	3	3.423	8
Serum-15min vs. Serum-45min	11.93	13.75	-1.822	0.7953	3	3	3.239	8
Serum-15min vs. Serum-60min	11.93	14.60	-2.675	0.7953	3	3	4.757	8
Serum-30min vs. Serum-45min	13.85	13.75	0.1035	0.7953	3	3	0.1840	8
Serum-30min vs. Serum-60min	13.85	14.60	-0.7496	0.7953	3	3	1.333	8
Serum-45min vs. Serum-60min	13.75	14.60	-0.8531	0.7953	3	3	1.517	8

Supplementary FIG A
One-way ANOVA analysis in reaction time of serum

Ordinary one-way ANOVA Multiple comparisons								
Number of families	1							
Number of comparisons per family	10							
Alpha	0.05							
Tukey's multiple comparisons test	Mean Diff.	95.00% CI of diff.	Significant?	Summary	Adjusted P Value			
1:40000 vs. 1:60000	-1.006	-7.384 to 5.372	No	ns	0.9833 A-B			
1:40000 vs. 1:80000	-1.775	-8.153 to 4.603	No	ns	0.8845 A-C			
1:40000 vs. 1:100000	0.7939	-5.584 to 7.172	No	ns	0.9931 A-D			
1:40000 vs. 1:120000	0.8798	-5.498 to 7.258	No	ns	0.9899 A-E			
1:60000 vs. 1:80000	-0.7683	-7.146 to 5.610	No	ns	0.9939 B-C			
1:60000 vs. 1:100000	1.800	-4.578 to 8.178	No	ns	0.8794 B-D			
1:60000 vs. 1:120000	1.886	-4.492 to 8.264	No	ns	0.8612 B-E			
1:80000 vs. 1:100000	2.569	-3.809 to 8.946	No	ns	0.6831 C-D			
1:80000 vs. 1:120000	2.654	-3.724 to 9.032	No	ns	0.6582 C-E			
1:100000 vs. 1:120000	0.08588	-6.292 to 6.464	No	ns	>0.9999 D-E			
Test details	Mean 1	Mean 2	Mean Diff.	SE of diff.	n1	n2	q	DF
1:40000 vs. 1:60000	11.23	12.24	-1.006	1.938	3	3	0.7343	10
1:40000 vs. 1:80000	11.23	13.01	-1.775	1.938	3	3	1.295	10
1:40000 vs. 1:100000	11.23	10.44	0.7939	1.938	3	3	0.5794	10
1:40000 vs. 1:120000	11.23	10.35	0.8798	1.938	3	3	0.6420	10
1:60000 vs. 1:80000	12.24	13.01	-0.7683	1.938	3	3	0.5607	10
1:60000 vs. 1:100000	12.24	10.44	1.800	1.938	3	3	1.314	10
1:60000 vs. 1:120000	12.24	10.35	1.886	1.938	3	3	1.376	10
1:80000 vs. 1:100000	13.01	10.44	2.569	1.938	3	3	1.874	10
1:80000 vs. 1:120000	13.01	10.35	2.654	1.938	3	3	1.937	10
1:100000 vs. 1:120000	10.44	10.35	0.08588	1.938	3	3	0.06267	10

Supplementary FIG B
One-way ANOVA analysis in dilution ratio of HRP-labeled antigen

Ordinary one-way ANOVA Multiple comparisons								
Number of families	1							
Number of comparisons per family	6							
Alpha	0.05							
Tukey's multiple comparisons test	Mean Diff.	95.00% CI of diff.	Significant?	Summary	Adjusted P Value			
15min vs. 30min	-0.3065	-5.119 to 4.506	No	ns	0.9967 A-B			
15min vs. 45min	-0.3268	-5.139 to 4.486	No	ns	0.9961 A-C			
15min vs. 60min	3.740	-1.072 to 8.553	No	ns	0.1365 A-D			
30min vs. 45min	-0.02033	-4.833 to 4.792	No	ns	>0.9999 B-C			
30min vs. 60min	4.047	-0.7657 to 8.859	No	ns	0.1026 B-D			
45min vs. 60min	4.067	-0.7453 to 8.879	No	ns	0.1007 C-D			
Test details	Mean 1	Mean 2	Mean Diff.	SE of diff.	n1	n2	q	DF
15min vs. 30min	14.80	15.11	-0.3065	1.503	3	3	0.2884	8
15min vs. 45min	14.80	15.13	-0.3268	1.503	3	3	0.3076	8
15min vs. 60min	14.80	11.06	3.740	1.503	3	3	3.520	8
30min vs. 45min	15.11	15.13	-0.02033	1.503	3	3	0.01914	8
30min vs. 60min	15.11	11.06	4.047	1.503	3	3	3.808	8
45min vs. 60min	15.13	11.06	4.067	1.503	3	3	3.827	8

Supplementary FIG C
One-way ANOVA analysis in reaction time for HRP-labeled antigen

Ordinary one-way ANOVA								
Multiple comparisons								
Number of families	1							
Number of comparisons per family	15							
Alpha	0.05							
Tukey's multiple comparisons test	Mean Diff.	95.00% CI of diff.	Significant?	Summary	Adjusted P Value			
TMB-5min vs. TMB-8min	-1.290	-5.430 to 2.850	No	ns	0.8930			A-B
TMB-5min vs. TMB-10min	-0.3704	-4.511 to 3.770	No	ns	0.9996			A-C
TMB-5min vs. TMB-12min	-0.3065	-4.447 to 3.834	No	ns	0.9998			A-D
TMB-5min vs. TMB-15min	-3.650	-7.791 to 0.4900	No	ns	0.0964			A-E
TMB-5min vs. TMB-20min	-3.520	-7.660 to 0.6206	No	ns	0.1144			A-F
TMB-8min vs. TMB-10min	0.9194	-3.221 to 5.060	No	ns	0.9717			B-C
TMB-8min vs. TMB-12min	0.9833	-3.157 to 5.124	No	ns	0.9625			B-D
TMB-8min vs. TMB-15min	-2.361	-6.501 to 1.780	No	ns	0.4385			B-E
TMB-8min vs. TMB-20min	-2.230	-6.370 to 1.910	No	ns	0.4951			B-F
TMB-10min vs. TMB-12min	0.06394	-4.076 to 4.204	No	ns	>0.9999			C-D
TMB-10min vs. TMB-15min	-3.280	-7.420 to 0.8604	No	ns	0.1554			C-E
TMB-10min vs. TMB-20min	-3.149	-7.290 to 0.9910	No	ns	0.1827			C-F
TMB-12min vs. TMB-15min	-3.344	-7.484 to 0.7965	No	ns	0.1433			D-E
TMB-12min vs. TMB-20min	-3.213	-7.354 to 0.9271	No	ns	0.1689			D-F
TMB-15min vs. TMB-20min	0.1306	-4.010 to 4.271	No	ns	>0.9999			E-F
Test details	Mean 1	Mean 2	Mean Diff.	SE of diff.	n1	n2	q	DF
TMB-5min vs. TMB-8min	11.67	12.96	-1.290	1.233	3	3	1.480	12
TMB-5min vs. TMB-10min	11.67	12.04	-0.3704	1.233	3	3	0.4250	12
TMB-5min vs. TMB-12min	11.67	11.98	-0.3065	1.233	3	3	0.3517	12
TMB-5min vs. TMB-15min	11.67	15.32	-3.650	1.233	3	3	4.188	12
TMB-5min vs. TMB-20min	11.67	15.19	-3.520	1.233	3	3	4.038	12
TMB-8min vs. TMB-10min	12.96	12.04	0.9194	1.233	3	3	1.055	12
TMB-8min vs. TMB-12min	12.96	11.98	0.9833	1.233	3	3	1.128	12
TMB-8min vs. TMB-15min	12.96	15.32	-2.361	1.233	3	3	2.708	12
TMB-8min vs. TMB-20min	12.96	15.19	-2.230	1.233	3	3	2.558	12
TMB-10min vs. TMB-12min	12.04	11.98	0.06394	1.233	3	3	0.07335	12
TMB-10min vs. TMB-15min	12.04	15.32	-3.280	1.233	3	3	3.763	12
TMB-10min vs. TMB-20min	12.04	15.19	-3.149	1.233	3	3	3.613	12
TMB-12min vs. TMB-15min	11.98	15.32	-3.344	1.233	3	3	3.836	12
TMB-12min vs. TMB-20min	11.98	15.19	-3.213	1.233	3	3	3.687	12
TMB-15min vs. TMB-20min	15.32	15.19	0.1306	1.233	3	3	0.1498	12

Supplementary FIG D
One-way ANOVA analysis in reaction time of TMB

- In Fig. 5A, it appears that only one serum per group was used for cross-reactivity analysis. For more robust results, it is recommended to use at least three samples per group for cross-reactivity studies. Utilizing multiple samples per group will improve the statistical power and reliability of the cross-reactivity analysis.

Response: We value your precious advice. In our previous specificity experiment, only one serum for each pathogen was used for cross-reactivity analysis in each group, which was indeed not rigorous and reliable. According to your suggestion, we re-conducted the cross-reactivity experiment, and three samples were used in each group for detection and verification, Fig. 5A has been corrected, and the corresponding description has also been updated and modified in line 140 and line 305.

FIG 5A Specificity of the DA_gS-ELISA.

(Notes: Data were visually displayed after re-conducted the cross-reactivity analysis with three samples)

- In line 137: Please quantify the sensitivity and specificity of the developed assay in percentage terms. Providing sensitivity and specificity as percentages is crucial because it offers a clear and precise measure of the assay's performance. Percentages allow for straightforward comparison with other assays and enable a better understanding of the assay's effectiveness in detecting true positives and negatives, which is essential for evaluating its diagnostic or analytical accuracy.

Response: Thank you for your valuable suggestion. The specificity analysis performed in our study was to simultaneously detect clinically positive serum samples of common swine pathogens, among them, FMDV was of highly similar symptoms to that of SVA infection. The OD₄₅₀ value was compared with the cut-off value to determine whether it was negative or positive, and the specificity was further evaluated by whether it was cross-reactive with other pathogens. Additionally, the sensitivity analysis involved in this study was to dilute positive serum samples by a continuous two-fold ratio ranging from 1:8 to 1:1024, which was finally evaluated by the limit of detection.

Although it is indeed more accurate and intuitive to present the data in the form of percentage quantification, I am afraid it is not suitable for the above evaluation method adopted in this study. In general, this study conforms to the evaluation principles of specificity and sensitivity, and does not affect the judgment of the final results.

FIG 5 Specificity and sensitivity of the DAgS-ELISA.

- In line 148, you mention that only 9 of the same serum samples were positive with DAgS-ELISA but negative with VNT. While this is understandable, a brief commentary on the underlying reasons for this discrepancy could be valuable. As scientific articles are among the best educational tools for younger scientists, providing such insights is important. Additionally, this presents an excellent opportunity to highlight the advantages of your developed ELISA test over the VNT test, emphasizing its potential superiority.

Response: Thank you for your valuable opinions Related supplements and explanations are made in the discussion section line 180-188.

- Also in line 30, you mention a kappa value of 0.78 to highlight the good agreement with the gold-standard VNT test. While this is an important point, it would also be valuable to emphasize that your developed test captures not only neutralizing antibodies but all SVA-specific antibodies, which results in higher sensitivity. Highlighting this aspect would further underscore the significance of your work.

Response: Thank you for your kind reminding. We have explored this in depth in the discussion section line 180-188.

- In line 284, additional information regarding who conducted the VNT test, where it was performed, and the methodology used would be beneficial. There is no information provided regarding the biosafety measures taken for VNT test. Providing these details will enhance the transparency and reproducibility of the study.

Response: Detailed procedures for VNT test are added in line 311-324.

- Also, it is unclear whether the serum samples used were inactivated. Providing these details is essential to ensure that appropriate biosafety protocols were followed.

Response: The serum samples used in the VNT test have been inactivated at 56°C for 30 min to ensure biosafety, and the detailed process is described in line 311-324.

Reviewer #2:

1. The introduction section has no title.

Response : Thank you for your kind reminding, the title of the introduction has been added to the line 50.

2. The information in the sentence between lines 89 and 91 must be referenced.

Response : The reference has been added and numbered.

3. Information should be briefly provided on the clinical course of the disease caused by Senecavirus A in pigs and why it is important for humans.

Response : Thank you for your valuable advice. The relative descriptions were mentioned in line 79-83.

Other changes

Besides the above correction, we tried our best to improve the manuscript and made some minor changes in the manuscript, which didn't list here but marked using 'track changes' in the marked manuscript. Beside the changes above, we have carefully checked the manuscript and not found any other questions.

We gratefully appreciate for Editors valuable comments and suggestions on our manuscript. We hope that the correction will meet with approval.

Re: Spectrum02043-24R2 (Development and Optimization of a Double Antigen Sandwich ELISA Detecting for Senecavirus A Antibodies Based on VP2 Protein)

Dear Dr. Jie Chen:

Your manuscript has been accepted, and I am forwarding it to the ASM production staff for publication. Your paper will first be checked to make sure all elements meet the technical requirements. ASM staff will contact you if anything needs to be revised before copyediting and production can begin. Otherwise, you will be notified when your proofs are ready to be viewed.

Sincerely,
Benjamin Liu
Editor
Microbiology Spectrum